# Urine Metabolome during Parturition

**DOI:** 10.3390/metabo10070290

**Published:** 2020-07-16

**Authors:** Federica Gevi, Alessandra Meloni, Rossella Mereu, Veronica Lelli, Antonella Chiodo, Antonio Ragusa, Anna Maria Timperio

**Affiliations:** 1Department of Biology and Ecology University of Tuscia, 01100 Viterbo, Italy; gevi@unitus.it (F.G.); v.lelli@unitus.it (V.L.); 2Neonatal Department, Obstetrics and Gynecology Unit, Azienda Ospedaliera Universitaria (AOU), 09124 Cagliari, Italy; gineca.ameloni@gmail.com (A.M.); rossellamereu@gmail.com (R.M.); antonellachiodo86@libero.it (A.C.); 3Department of Obstetrics and Gynecology, Ospedale San Giovanni Calibita, Fatebenefratelli, Isola Tiberina, Via di Ponte Quattro Capi, 39, 00186 Roma, Italy; antonio.ragusa@gmail.com

**Keywords:** urine metabolomic profile, out of labor, in labor, dilating phase, estrogens, conjugated estrogens, amino acid

## Abstract

In recent years, some studies have described metabolic changes during human childbirth labor. Metabolomics today is recognized as a powerful approach in a prenatal research context, since it can provide detailed information during pregnancy and it may enable the identification of biomarkers with potential diagnostic or predictive. This is an observational, longitudinal, prospective cohort study of a total of 51 serial urine samples from 15 healthy pregnant women, aged 29–40 years, which were collected before the onset of labor (out of labor, OL). In the same women, during labor (in labor or dilating phase, IL-DP). Samples were analyzed by hydrophilic interaction ultra-performance liquid chromatography coupled with mass spectrometry (HILIC-UPLC-MS), a highly sensitive, accurate, and unbiased approach. Metabolites were then subjected to multivariate statistical analysis and grouped by metabolic pathway. This method was used to identify the potential biomarkers. The top 20 most discriminative metabolites contributing to the complete separation of OL and IL-DP were identified. Urinary metabolites displaying the largest differences between OL and IL-DP belonged to steroid hormone, particularly conjugated estrogens and amino acids much of this difference is determined by the fetal contribution. In addition, our results highlighted the efficacy of using urine samples instead of more invasive techniques to evaluate the difference in metabolic analysis between OL and IL-DP.

## 1. Introduction

The mechanisms involved in the physiological onset of human labor are complex and still poorly understood. Timing is critical for both maternal and infant survival, and it is also associated with newborn maturity [1]. Therefore, the diagnosis of labor is one of the most important aspects in pregnancy care. Nevertheless, the factors triggering the onset of labor are still unknown. Labor usually starts within two weeks before or after the estimated date of delivery. Some women experience very clear signs of labor, while others do not. In addition, women experience labor in different ways, and signs may occur either before labor starts or during labor.

As stated in 1996 by the World Health Organization (WHO) (Geneva, Switzerland), labor diagnosis is the most important aspect in the management of labor [2]. When the initial diagnosis is not correct, the subsequent management has a high probability of being problematic. Wrong diagnosis (15%) may lead to inappropriate labor care, with consequences such as intervention for labor dystocia or, the other way around, prolonged labor, which can lead to an increase in operative deliveries and/or fetal distress, with possible adverse maternal-neonatal outcomes. In addition, false labor or Braxton-Hicks contractions can occur, resulting in hospitalizations and excessive medical treatment, with psychological consequences for mothers and a waste of economic resources. Several reports show an increasing incidence of hospitalization for false labor [3,4,5,6]. Although several authors have recognized the importance of correct labor diagnosis, and many efforts have been made to establish the correct criteria for defining labor, at present, a correct and useful tool allowing an accurate diagnosis of its onset is not available.

Several studies have been addressed towards understanding the hormonal mechanism of labor onset in animals, using uterine myocytes collected and cultured in vitro, maternal serum, placenta, and fetal membranes and decidua. Although interesting results were obtained, it is difficult to extrapolate animal model results to humans, due to the differences regarding pregnancy between species [7,8,9,10]; these mediums are in any case more difficult to obtain than urine.

Metabolomics today is recognized as a powerful approach for dealing with several aspects of modern research. It is an analytical strategy for measuring the metabolic responses of living systems to various stimuli and it suited to both targeted and untargeted metabolite profiling [11]. Several metabolomic studies that can be further listed employ both high-resolution proton nuclear magnetic resonance (1H NMR), which provides an overview of more abundant metabolites at micromolar levels, structural information and high reproducibility, and mass spectrometry (MS), which offers higher sensitivity at nanomolar levels and the possibility of optimized targeted analysis, particularly when combined with gas chromatography (GC) or liquid chromatography (LC). Metabolomics currently allows us to separate the population of women in labor from women not in labor by analyzing urinary metabolites [11,12,13].

Furthermore, by investigating the details, it also allows us to separate the populations of women with intact membranes in labor or not in labor [13]. Hence, the aim of this study was to evaluate the differences between two times of labor defined out of labor (OL) and in labor or dilating phase (IL-DP) through a procedure able to detect a wide variety of metabolites from maternal urine samples taken from the same woman for a more correct labor diagnosis. To ensure broad metabolite detection coverage on urine sample, we employed hydrophilic interaction chromatography (HILIC)–LC-electrospray ionization (ESI)-MS, a technology particularly suitable to separate sample and complex mixtures in biological fluids [14,15]. Applying this experimental approach, metabolic differences were able to discriminate between the two phases OL and IL-DP, highlighting the metabolites most involved in the discrimination.

## 2. Results

Our untargeted metabolomic results confirm the retrospective clinical diagnosis of labor, the analysis evidenced clusters of metabolites involved in the labor condition able to discriminate between urine samples collected out of labor (OL) and during labor (in labor or dilating phase IL-DP).

### 2.1. Urine Metabolomic Study of Labor

The representative HILIC-MS chromatograms from positive and negative ion modes of human urine from the women at different time of labor was compared. Using the optimized HILIC-MS analysis protocol and subsequent processes, such as baseline correction, peak deconvolution, alignment, and normalization, we obtained a three-dimensional matrix, including data file name, retention time exact mass pair, and normalized peak areas. Although some differences could be visually noted among the three sets of the detail in the chromatogram, more subtle changes could be found using a pattern recognition approach, such as PLS-DA. Typically, the metabolic profiles of OL and IL-DP are compared with the aim of identifying spectral features, and ultimately metabolites, which discriminate the classes.

### 2.2. Analysis of Metabolic Pattern

Using our metabolomics platform, the statistically important metabolites were studied. PCA was used first to investigate general interrelation between groups (Appendix A (Appendix A), Figure 1). PLS-DA was used to maximize the difference of metabolic profiles between OL and IL-DP and facilitate the detection of metabolites consistently present in the biological samples. Data were standardized using Sum-Autoscaling. The urinary metabolomics of two sample sets OL and IL-DP are largely distinguishable, and trajectory analysis of score plots (3D) on the tridimensional PLS-DA showed clear segregation, which suggests that urinary biochemical differences significantly occurs in the two groups, depicting the first three principal components (PC), which together explain 49.4% of the total variance (accuracy, Q2 and R2 data are shown in Appendix A (Appendix A)). Approximately 10,000 peaks per sample were obtained with reference to the KEGG database; among them, 356 metabolites were analyzed more precisely and identified.

Metabolite profiling focuses on the analysis of a group of metabolites related to a specific metabolic pathway in biological states. It is a web-based tool that combines the results from powerful pathway enrichment analysis in the condition under study. Metaboanalyst, a directed graph, uses the high-quality KEGG pathway database as the backend knowledge base. Consequently, potential targets of metabolic pathway analysis (Figure 2) (Impact-value ≥ 0.10) with Metaboanalyst revealed that the metabolites that were identified together are important for the host response to labor, the top two metabolic pathways of importance, including steroid hormone biosynthesis and amino acid metabolism, were perturbed (Appendix A (Appendix A)).

Given the relevance of steroid hormones in gravidancy, the hormone pathway was assessed in greater detail.

### 2.3. Steroid Hormone

Table 1 shows the detailed results from the pathway analysis. Since many pathways were tested at the same time, the statistical *p*-values from enrichment analysis were further adjusted for multiple testings. From this table, 99 compounds belonged to the steroid hormone, with a maximum hits value of 39, which means that 39 hormones were found in our analysis, as emerged in Metaboanalyst. Urinary estrogen concentrations (estradiol, estrone and estriol) excreted as free or conjugated with glucoronide and/or sulphate, increased in IL-DP (Table 1) and were detected through positive or negative ionization mode. Appendix A (Appendix A) represents the alignment of three technical replicates of 29 samples in OL and 22 samples in IL-DP. Using a typical UPLC system coupled to a MS scanning at 1 Hz, 20 scans were performed, with a very high chromatographic reproducibility [16]. To manually assess alignment quality, electro-ion chromatograms (EICs) of some common metabolites were analyzed. Appendix A (Appendix A) shows estrogen EICs, where A stands for samples collected in OL and B for those collected in IL-DP. The size of the circle at the top of each EIC represents an auto-generated quality score, with larger circles denoting higher quality. They were identified as conjugated pathways of estrone and conjugated estradiol and among them, estrone sulphate and estradiol 17-beta-3-gluconoride were the conjugated forms primarily increased in the IL-DP. The excretion of these hormones in the conjugated form reached a maximum in the IL-DP (Table 1).

### 2.4. Amino Acid

Appendix A (Appendix A) show most of the amino acids found in urine collected in different phases. All the amino acids found in our analysis, as detected by mass spectrometry, are summarized in Table 2. Among them, the amount of serine, alanine, phenylalanine and tyrosine decreased, whereas valine, histidine, arginine, cysteine, glutamate, glutamine, leucine, lysine, isoleucine and threonine increased in the urine samples from IL-DP compared with those detected in OL, ranging from a minimum of 11% for phenylalanine to a maximum of 85% for glutamate. Upon alignment and matching of mass spectra chromatogram and feature detection throughout the whole retention time range, scatter plot analyses, including those features present in at least all replicates of one group in a statistically significant manner (*p* < 0.05, *t*-test), were performed by comparing results from OL samples with those from IL-DP.

## 3. Discussion

Much remains unknown about the mechanisms of the physiological onset of labor in humans. Processes that lead to readiness for labor, birth, and post-partum transitions are coordinated at term labor between mother and baby, whose maturity ultimately determines timing, according to current understanding.

The results of this study highlighted the possibility of distinguishing between two different metabolic profiles from urine samples associated with OL and IL-DP by a fast and easy method, thus revealing the metabolic profile of labor through urine samples, which are easy to collect, available in large volumes, and are one of the best biological fluids in metabolomics, since they are mostly free from interfering proteins or lipids. Therefore, the possibility of using urine samples available over time and in large amounts allowed us to perform a longitudinal study by collecting samples just before and into labor, elucidating for the first time, to our knowledge, that metabolites are implicated in the late phases of pregnancy, and could be considered predictive of imminent labor. Hormones and amino acids were the compounds that underwent major changes between the two groups, as shown in Table 1 and Table 2. Recently, it has been reported that maternal estrogen levels increase because the baby’s maturing adrenal gland produces increasing amounts of the estrogen precursor dehydroepiandrosterone-sulfate (DHEAS) in the four to six weeks before birth. DHEAS is converted to estrogen in the placenta and then enters the mother’s circulation. Changes in the ratios of estrogen to progesterone, of the estrogen subtypes estriol and estrone, and/or of progesterone receptors A and B may all be involved in uterine activation, and possibly in the physiologic onset of labor [1]. Our data allowed us to determine that during IL-DP, the excretion of estrogens in their conjugated form reached a maximum. These results are in agreement with some authors who found that estrogen sulphates exhibit a much longer half-life than the parent estrogens [17,18], thus explaining their prevalence over other estrogens in urine. In the past, it was assumed that sulphate and glucuronide conjugation of estrogens represented a pathway resulting in less active, more polar and more readily excreted estrogenic compounds. It is now appreciated, however, that estrone sulphate is the most abundant circulating estrogen, with concentrations approximately 10-fold higher than unconjugated estrone. In our case, the comparison of two pregnancy phases very close to each other revealed that all hormones were conjugated and increased during IL-DP. According to Fabregat [19], estradiol and estrone exhibit similar behaviors, increasing approximately 100-fold between basal levels and the levels present in the third trimester. These results are in agreement with those previously reported in the literature [17,18,19,20,21]. Since both estradiol and estrone are produced in the placenta and can be interconverted by the enzyme 17-hydroxysteroid dehydrogenase [22], a similar behavior is expected. However our results showed that although they are all secreted as sulphate or glucuronide conjugates, sulphation and desulphation of estrogens may well represent an endogenous system important in the regulation of biologically active steroid hormones in target tissues, as well as transport among the fetus, uterus and placenta; this is supported by an increasing body of evidence. Specifically, it is currently hypothesized that inactive estrone sulphate is transported to target tissues via the circulatory system, taken into target cells most likely by organic anion transporters, enzymatically hydrolyzed to estrone by intracellular membrane-bound steroid sulfatase (arylsulfatase C), and hydroxylated to active 17-estradiol via catalysis by 17-hydroxysteroid dehydrogenases [20]. We might therefore postulate a higher excretion of such conjugates, resulting in increased hormone synthesis during this phase by fetal-placental production, leading to a progressive increase in maternal circulating levels. Regarding estriol, it is produced during pregnancy at increasing concentrations during pregnancy itself. Its synthesis is mainly feto-placental. It passes through the placenta into the maternal circulation and is rapidly converted into glucuronide and sulphate derivatives to facilitate its excretion. Levels of estriol are used to determine whether the pregnancy is progressing normally [23]. Using our technique, it was possible to quantify estriol levels at nano-molar concentrations during OL and IL-DP pregnancy phases. Up until this point, estriol circulates in the blood as both free unconjugates and conjugates, and it is the only estrogen for which the diversified dosage is currently used: the unconjugated form and the total conjugated form (for the determination of the latter, preliminary enzymatic hydrolysis is necessary). The production of estriol during normal pregnancy is constantly increasing: low levels of this steroid may, however, be observed in the case of lack of the placental enzyme sulfatase, although the fetus develops normally in this condition [22]. Estriol precursors are mainly produced by the adrenal fetal gland, explaining the existence of a circadian rhythm in the concentration of this steroid in the maternal serum characterized by higher levels of up to 8 pm, with a minimum at 8 am. Estriol and its conjugate have been widely investigated for monitoring fetal function. Using our method, we were able to detect estriol, estrone and estradiol (sulphate and glucuronide) in their free and conjugated forms without preliminary enzymatic hydrolysis, it may serve to control the circadian cycle in the urine without taking a blood sample. It is remarkable that a single chromatographic run, in several minutes and in an easily available sample, could determine all the components of conjugative pathways for estrone, estradiol and estriol. Appendix A shows the metabolism of estradiol in urine; using our method, all compounds involved could be detected simultaneously. The increase in urinary excretion of amino acids during pregnancy is well known, and many efforts have been done to measure them. HILIC MS could detect all the amino acids present in urine, it was unaffected by the presence of amino acids in combined forms or homologues, and notably, it facilitated the identification of previously unknown constituents. In general, pregnancy is usually accompanied by selective amino aciduria. Glucose and amino acids may not be absorbed efficiently. Hence, glycosuria and amino aciduria may develop during normal gestation [23]. An increased placental transfer of amino acids, favoring nitrogen conservation for fetal growth, can lead to diminished circulating levels of many amino acids in plasma, a condition known as hypoamino acidemia. This should also lead to an underuse of many amino acids in maternal gluconeogenesis, particularly later in gestation, and alternative substrates such as glycerol are used, followed by accelerated fat breakdown or marked changes in protein metabolism, favoring nitrogen conservation and active transfer of amino acids to the fetus [24].

Among the glucogenic amino acids, serine was the amino acid that underwent the greatest decrease (59%) between women in OL compared with those in IL-DP, while glutamate was the amino acids that underwent the greatest increase (Table 2) between women in OL compared with those in IL-DP. Serine influences placental-fetal development and glutamate plays an important role in fetal growth and development. For these reasons, we believe that these two glucogenic amino acids possess key roles during pregnancy and active labor. Serine is known for its role in intermediary metabolism as a source of one-carbon pool for nucleotide biosynthesis, as a precursor of glycine and glucose, and as a contributor to cysteine biosynthesis. A unique serine-glycine cycling between liver and placenta has been demonstrated in the sheep fetus [25]. The transport of serine across the placenta in human pregnancy remains uncertain because of umbilical artery and vein concentration differences, as no net transfer is suggested in some studies [26,27], and significant transfer is suggested in others [27,28]. In our study, we detected the amount of glycine, a serine precursor, below the meaningful minimum in all samples, independently on the clinical condition; thus, we hypothesized that glycine was taken up by the fetus from the placenta and converted to serine in the fetal liver as previously described [29,30]. The importance of serine metabolism during fetal development is underscored by its role as a precursor of nucleotide biosynthesis during the rapid growth phase in neonatal rats [31,32] and as a source of the one-carbon pool for nucleotide synthesis and during cell proliferation [33]. A paper on serine metabolism in human pregnancy concluded that plasma serine levels were lower during pregnancy than in non-pregnant women, and that a significant decrease in serine turnover is evident in late gestation [33]. Our findings agree with the above previously published literature. Therefore, serine detection could be both diagnostic for the success of pregnancy and can also be a candidate biomarker for diagnosis of labor. Although no net release of serine by the liver has been demonstrated, serine is metabolized in the liver in both the adult and the fetus [27,28]. Nevertheless, our results were derived from urinary excretion, and thus they contained serine in all its forms, as well as oxidized serine. Indeed, serine can be oxidized either via pyruvate or by conversion to glycine and oxidation by the glycine cleavage system: serine and glycine have been shown to have a unique metabolism in the fetus and placenta [32]. The decrease in serine in advancing gestation may be related to the down-regulation of transaminations to conserve nitrogen, as was previously observed, and as also reported by Marcos et al. [22].

Our results showed an increase in glycogenic amino acids as well as ketogenic amino acids in urine samples. An exchange of glutamine–glutamate occurs between the placenta and fetal liver [34]. Glutamine enters the pregnant uterus from the maternal circulation, is released by the placenta into the umbilical circulation, and is taken up by the fetal liver [35,36]. The net fluxes of glutamine into the pregnant uterus and into the fetus are large in comparison with the net fluxes of other amino acids [35,36]. This evidence is also confirmed by our data: glutamate reached 85% of excretion during IL-DP. We might also state that the uptake of glutamine is almost all released as glutamate by the fetal liver, although it is a gluconeogenic amino acid. Similarly, the fetal hepatic uptake of glutamine appears to be greater than that of any other amino acid [37]. Our data showed a significant increase of these two amino acids in IL-DP, particularly glutamate, which increased by a factor of nearly 10. Of the amino acids we found, glutamate is the most excreted and therefore also the main amino acid produced. The glutamate entering the placental tissues is largely oxidized [38]. In exploring the physiological meaning of these observations, it is important to remember that the fetus has a high rate of oxidative amino acid metabolism because the placenta delivers to the fetus amino acid nitrogen in excess of what is needed for nitrogen accretion [38,39]. Therefore, reciprocal glutamate–glutamine fluxes between the placenta and fetal liver may serve multiple functions, such as delivery of glutamine amide nitrogen to the fetal liver, release by the fetal liver of amino acid carbon that cannot be converted to glucose and linking of placental oxidative metabolism and NADPH production for steroidogenesis to fetal liver metabolism of glutamine [40]. The high rate of conversion of fetal plasma glutamine to fetal plasma glutamate by the fetal liver is related to the fact that there is virtually no gluconeogenesis during fetal life to this glutamine and glucose metabolism play important and unique roles during fetal development. The absence of gluconeogenesis is functionally advantageous for the fetus because maternal glucose is transported into the fetal circulation by facilitated diffusion, which requires the maintenance of a transplacental glucose concentration gradient. Glucose production by the fetal liver would reduce the gradient and block the acquisition of maternal glucose [41]. Instead of glucose, the fetal liver delivers into the fetal circulation glutamate, which is subsequently oxidized by other fetal organs and the placenta [42]. This evidence could explain the high amounts of glutamate we found.

## 4. Materials and Methods

### 4.1. Subjects and Urine Collection

Fifty-one samples were collected using sterile containers from 15 healthy pregnant women, nulliparous and multiparous, 29–40 years of age, recruited from the Department of Obstetrics and Gynecology, Azienda Ospedaliera Universitaria (AOU) in Cagliari, Italy, from November 2013 to December 2014. Urine samples were then frozen, shipped in dry ice, and stored at −80 °C continuously until analysis. The study conformed to the principles outlined in the Declaration of Helsinki, and was approved by the ethics committee of the AOU, Cagliari, Italy. All patients gave their written informed consent prior to their participation in the study. The inclusion criteria were as follows: (1) enrolment before labor onset; (2) low-risk pregnancies; (3) full-term, cephalic, single fetuses, appropriate for gestational age; (4) labor not complicated by fever and/or meconium-stained amniotic fluid. Gestational age at enrollment and delivery was between 38 weeks and 5 days and 41 weeks and 2 days of gestation. Gestational age was based on the date of the last menstrual period and confirmed by ultrasound scan. Specimen collection started before labor onset, and the same woman collected serial specimens during labor, depending on maternal compliance, since some patients were unable to urinate or to collect urine during labor. Among the 51 collected samples, 29 samples were collected before the onset of labor (OL) and 22 during labor (IL) in the dilating phase (IL-DP). Samples were considered as belonging to the labor group when collected from women with regular painful uterine contractions associated with cervical dilatation ≥ 5 cm [13,14]. Each sample was identified with an alpha numeric code, in order to know the precise characteristics of each of them, thus providing the information important for an accurate retrospective labor diagnosis [1].

### 4.2. Metabolite Extraction aHILIC-UHPLC Separation

Metabolite extraction and separation was performed according to Gevi et al. 2017 [43].

### 4.3. Mass Spectrometry

Mass Spectrometry (MS) analysis was carried out on an electrospray hybrid quadrupole time-of-flight instrument MicroTOF-Q (Bruker-Daltonik, Bremen, Germany) equipped with an ESI ion source, as previously described [44], with some modifications; ESI capillary voltage was set at 4500 V (−) ion mode, nebulizer set at 27 psi, and the nitrogen drying gasset to a flow rate of 6 L/min. Dry gas temperature was set at 200 °C.

### 4.4. Data Elaboration and Statistical Analysis

Replicates were exported as mzXML files and processed through MAVEN.52 (available at http://genomics-pubs.princeton.edu/mzroll/index.php?show=index) [16,45]. Mass spectrometry chromatograms were elaborated for peak alignment, matching and comparison of parent and fragment ions, and tentative metabolite identification (within a 10-ppm mass deviation range between observed and expected results against the imported Kyoto Encyclopedia of Genes and Genomes (KEGG) database). Multivariate statistical analyses were performed on the entire metabolomics data set using the MetaboAnalyst 4.0 software (http://www.metaboanalyst.ca) [46], which also provided an overview of data variance structure in an unsupervised manner and produced scatter plots. Partial least squares discriminant analysis (PLS-DA), which defines a predictive model that describes the direction of the maximum covariance between a dataset (X) and class membership (Y), was then used to maximize the difference in metabolic profiles between cases and controls [47,48]. PLS-DA was performed using the Excel add in Multibase package (Numerical Dynamics, Japan; http://www.numericaldynamics.com/) by applying partial signal correction on the metabolite concentrations shifted, sum transformed, centered, and autoscaling to unit variance. For case-control contrasts of single urinary metabolites, significance threshold was held at a nominal *p* < 0.05 with no correction for multiple testing, because (a) differences in single metabolite concentrations were tested only following significant differences in pathway enrichment were detected, (b) intra-pathway variability of single metabolites is non-independent, and (c) different metabolic pathways are also not fully independent, as some metabolites fall into more than one pathway. Potential markers of interest were extracted from VIP plots that were constructed from the PLS DA analysis, and markers were chosen based on their contribution to the variation and correlation within the data set.

## 5. Conclusions

Urine typically contains a wide range of metabolites. However, we focus our attention on pregnancy-specific metabolites detectable in maternal urine at OL and IL-DP. The increased excretion during labor of conjugated estrogens observed in our study may confirm the coordinated role played between the fetus and the mother during labor. Therefore, labor is confirmed not to merely be the mechanical event linked to increased uterine contraction, but rather a more complex process.

## Figures and Tables

**Figure 1 metabolites-10-00290-f001:**
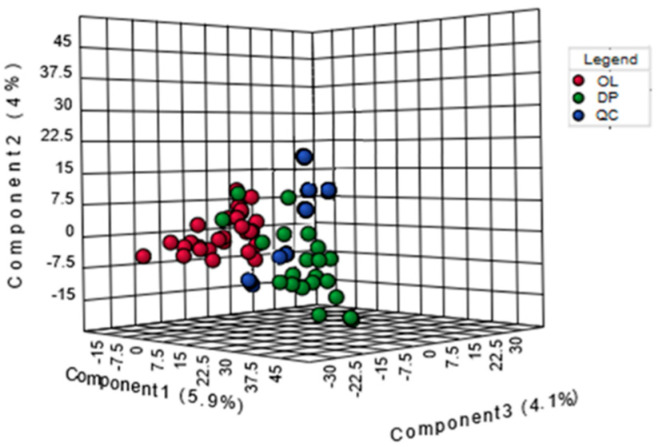
PLS-DA 3D plot based on normalized and mean-centered data. Each data point represents the metabolome of a single individual. In red are expressed sample collected in OL. In green are expressed samples collected in IL-DP. In blue are expressed QC.

**Figure 2 metabolites-10-00290-f002:**
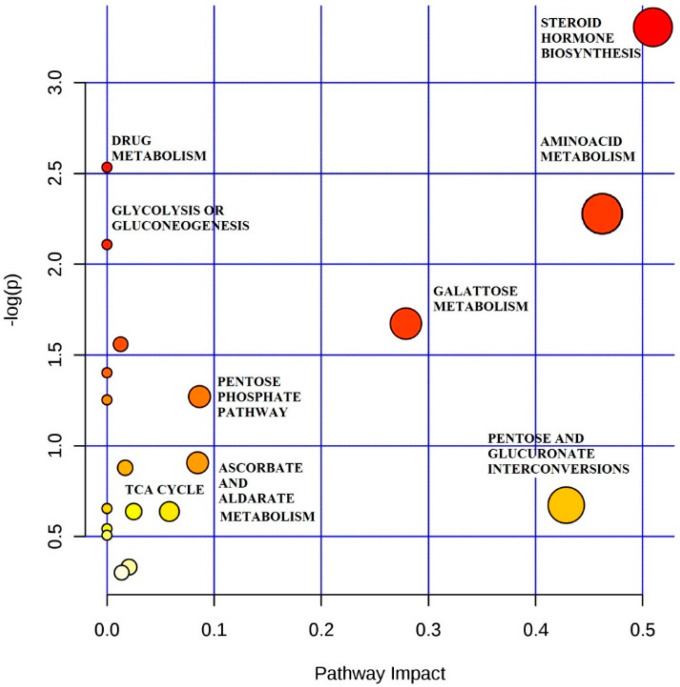
Metabolic Pathway Analysis (MetPA). All the matched pathways are displayed as circles. The color and size of each circle are based on the *p*-value and pathway impact value, respectively. The graph was obtained by plotting on the *y*-axis the −log of *p*-values from the pathway enrichment analysis and on the *x*-axis the pathway impact values derived from the pathway topology analysis.

**Table 1 metabolites-10-00290-t001:** Estrogen amount extracted from the urine in the two stages of pregnancy (OL and IL-DP). The table refers to the relative concentration of the metabolites calculating as a percentage of the total hormones by comparing the intensity of deconvolution of each hormone.

Estrogens	Molecular Weight	Detection Mode	Out of Labor	Active Phase	Retention Time	%
3 hydroxy2-methyl-1H-quinolin-one	175.05	Positive	9300	95,000	7:00	3% ↑
19 chloro19-Chloro-3beta-hydroxyandrost-5-en-17-one = dehydroepiandrosterone	365.1695	Positive	20000	39,000	5:0035	95% ↑
Androst-5-ene-3beta,17beta-diol = androsterone	290.1736	Positive	28,000	22,000	7:50	21% ↓
Androsterone	290.1736	Positive	28,000	22,000	7:50	21% ↓
Dehydroepiandrosterone Sulfate	369.0992	Positive	39,000	31,00	5:05	92% ↓
Dehydroepiandrosterone	288.1873	Positive	27,000	26,000	7:30	3% ↓
Pregnanediol	321.2145	Positive	8000	6000		25% ↓
3-Hydroxy-1-methylestra-1,3,5(10),6-tetraen-17-one	283.1551	Negative	30,000	11,000	5:50	63% ↑
Tetrahydrocortisone	365.1592	Positive	140,000	16,000	6:30	88% ↓
Estrone 3 sulfate	351.1085	Positive	13,800	13,000	11:00	>100% ↑
Estrone gluconoride	447.1790	Positive	79,000	10,000	12:30	87% ↓
Estradiol 17 beta 3 gluconoride	449.1945	Positive	14,000	20,000	12:00	42% ↑

**Table 2 metabolites-10-00290-t002:** Amino acid amounts extracted from the urine in the two stages of pregnancy (OL and IL-DP). The table refers to the relative concentration of the metabolites, calculated as a percentage of the total amino acids, by comparing the intensity of deconvolution of each amino acid.

Glucogenic Amino Acid	Molecular Weight	Detection Mode	Out of Labor	Active Phase	Retention Time	%
Ser	105.09	Negative	3400	1400	12:02	59% ↓
Val	117.15	Positive	43,000	66,000	6:00	35% ↑
His	155.15	Negative	72,000	94,000	13:04	23% ↑
Arg	174.20	Negative	1542	1742	14:00	11% ↑
Cys	121.16	Negative	1249	2383	15:50	47% ↑
Ala	89.09	Positive	3400	1600	20:02	47% ↓
Glu	147.13	Negative	1800	12,000	4:02	85% ↑
Gln	146.14	Positive	61,000	82,000	21:00	25% ↑
**Ketogenic Amino Acid**	**Molecular Weight**	**Detection Mode**	**Out of Labor**	**Active Phase**	**Retention Time**	**%**
Leu	131.17	Positive	16,000	26,000	3:50	38% ↑
Lys	146.19	Negative	3900	9700	19:00	60% ↑
**Glucogenic and Ketonic**	**Molecular Weight**	**Detection Mode**	**Out of Labor**	**Active Phase**	**Retention Time**	**%**
Ile	131.17	Positive	16,000	26,000	12:00	38% ↑
Thr	119.12	Positive	5600	6800	18:00	18% ↑
Phe	165.19	Positive	180,000	160,000	17:00	11% ↓
Tyr	181.19	Positive	170,000	150,000	5:00	12% ↓

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
