# Peer review of "Urine Metabolome during Parturition"

_metabolites, 2020, doi:10.3390/metabo10070290_

Round 1

Reviewer 1 Report

The manuscript „Urine metabolome during parturition“covers important topic. Unfortunately it has several important issues.

-              I found analyzing QCs very 15th sample (chapt. 4.2.) very problematic! How the QC samples were used and quality control conducted? If QC process is not correct - this is exclusion criterion in my view.

-              Figures 3 – 5 do not provide too much for the reader and are abundant.

-              Table 1 and 2 require improvement (more precise explanation what is calculated)

-              the authors mentioned use of VIP from PLSDA, ANOVA and scatter plots – none of it is shown.

-              In Fig. 1: component 2 explains 4.3 and component 3 4.6%, respectively. It seems that numbers follow projection used for Fig. 1.

-              Ch 2.2. L 95 – PCA is not a tool for outlier detection

-              Ch 2.2. L 109: why authors used an impact value of 0.10?

-              Table 1S?

-              Fig. 2: pathways  below - log(0.05)=1.3 (even higher if p-value correction is used ) are not important

-              Second sentence in 2.3 requires attention

-              2.3. Chap. L132 “cmpd”?

-              2.3. Chap. L146. Sentence starting w. “Estrogen...” does not fit here,

-              2.4. Chap. L174 correction for No of metabolites is missing

-              4.5. chap L405 - 411: reasoning a) (L405) is not substantiated and authors should correct for No of metabolites.

-              4.5. chap Authors do not provide data/results for justifying the last sentence.

Author Response

Dear  Editor of Metabolites

Manuscript No. metabolites-821575

Title: Urine metabolome during parturition

Authors: Federica Gevi, Alessandra Meloni, Rossella Mereu, Veronica Lelli, Antonella Chiodo, Antonio Ragusa, Anna Maria Timperio.

First of all, we would like to thank you and the Reviewers for the positive and encouraging comments on the manuscript.

Please, find enclosed a point by point reply to each one of the Reviewers’ comments. We sincerely hope that this deeply revised version of the manuscript will succeed in meeting the Reviewers’ expectations. All changes are highlighted using track changes function in Microsoft word.

Kind regards

Prof. Timperio Anna Maria

Department of Ecological and Biological Sciences (DEB)

University of Tuscia

Largo dell’Università

01100 Viterbo, ITALY

Phone: +39 0761357100 (direct)

Fax: +39 0761357179 (dept.)

E-mail: timperio@unitus.it 

Reviewer #1

The manuscript “Urine metabolome during parturition“ covers important topic. Unfortunately it has several important issues.

We thank reviewer #1 for these helpful comments and we reviewed the manuscript following his suggestions.

Reviewer #1 I found analyzing QCs very 15th sample (chapt. 4.2.) very problematic! How the QC samples were used and quality control conducted? If QC process is not correct - this is exclusion criterion in my view.

Authors' reply: The authors apologize for the inconvenient caused. In reality there was an error in typing,  the QC analyses were repeated every 5 runs and in the test we have at least 10 QC points to show in the figure. The QC was done by taking 10 ul from each sample and analyzed every 5 samples this allows us to obtain data that justify the stability of the run. We enclose a new figure 1 of the PLS-DA 3D in which we also show the QC.

Reviewer #1  Figures 3 – 5 do not provide too much for the reader and are abundant.

Authors' reply: We thank the reviewer to bring these to our attention. The images quality has been improved  and they have been uploaded in supplementary material files

Reviewer #1 Table 1 and 2 require improvement (more precise explanation what is calculated

Authors' reply Tables 1 and 2 refer to the relative concentration of the metabolites taking into consideration the variation between OL and DL, calculating as a percentage of the total hormones and amino acid by comparing the intensity of deconvolution of each hormone and amino acid. In the new version we have improved the figure captions

Reviewer #1 the authors mentioned use of VIP from PLSDA, ANOVA and scatter plots – none of it is shown.

Author’s reply : This part refers to additional analyses we have done but which we decided not to present because they are too redundant and irrelevant for the manuscript therefore we have removed the sentences  in the text. 

Reviewer #1 In Fig. 1: component 2 explains 4.3 and component 3 explains 4.6%. It seems that numbers follow projection used for Fig. 1.

Author’s reply:  We have uploaded a new figure 1 containing the QC and then rewritten the various component values.

Reviewer #1 Ch 2.2. L 95 – PCA is not a tool for outlier

Author’s reply  thank you for the Reviewer #1 suggestion we modified the text

Reviewer #1 Ch 2.2. L 109: why authors used an impact value of 0.10?

Author’s reply we use an Impact value > 0.10 as a suggestion of MetaboAnalyst program which defines a value greater than 0,10 as significant.  Furthermore other authors give the same value

Pang, Z., Chong, J., Li, S. and Xia, J. (2020) MetaboAnalystR 3.0: Toward an Optimized Workflow for Global Metabolomics. Metabolites 10(5) 186 (R code underlying MetaboAnalyst web server)

Urine metabolomics analysis for biomarker discovery and detection of jaundice syndrome inpatients with liver diseaseXijun Wang*, Aihua Zhang, Han Ying, Ping Wang, Hui Sun, Gaochen Song, Tianwei Dong, Ye Yuan, Xiaoxia Yuan,Miao Zhang, Xie Ning*, He Zhang, Hui Dong, Wei Dong

Urine Metabolomics Analysis for Biomarker Discovery and Detection of Jaundice Syndrome in Patients With Liver Disease S Xijun Wang, Aihua Zhang, Ying Han‡, Ping Wang, Hui Sun‡, Gaochen Song‡,Tianwei Dong‡, Ye Yuan‡, Xiaoxia Yuan‡, Miao Zhang‡, Ning Xie‡§, He Zhang‡, Hui Dong‡, and Wei Dong‡

Reviewer #1 Table 1S?

Author’s reply  We apologize unfortunately there was a problem in loading table 1S in the new version of manuscript we reloaded the table. 

Reviewer #1 Fig. 2: pathways  below - log(0.05)=1.3 (even higher if p-value correction is used ) are not important  

Authors' reply We completely agree with the reviewer #1 infact pathways  below - log(0.05)=1.3 are not important for this reason we considered for our analyses both parameters: -Log(P) and pathway impact under 2 and 0,4 respectively.

Reviewer #1 2.3. Chap. L132 “cmpd”?

Authors' reply: “cmpd” has been correct to “compound”

Reviewer #1 2.3. Chap. L146. Sentence starting w. “Estrogen...” does not fit here,

Authors’  reply : the sentence has been removed               

Reviewer #1 2.4. Chap. L174 correction for No of metabolites is missing.

 Authors’  reply: the correction used for No of metabolites  is a t-test .

Reviewer #1 4.5. chap L405 - 411: reasoning a) (L405) is not substantiated and authors should correct for No of metabolites -4.5. chap Authors do not provide data/results for justifying the last sentence.

Authors’  reply :   This part refers to additional analyses we have done but which we decided not to present because they are too redundant and irrelevant for the manuscript therefore we have removed the sentences  in the text. 

Reviewer 2 Report

The manuscript titled "Urine metabolome during parturition" submitted to Metabolites, describes a very interesting and useful approach using metabolomics to evaluate the differences between two times of labor defined out of labor (OL) and in labor or dilating phase (IL-DP). The presented results confirm the retrospective clinical diagnosis of labor evidenced clusters of metabolites involved in labor condition able to discriminate between urine samples collected out of labor (OL) and during labor (in labor or dilating phase IL-DP).  The study seems to be properly designed and authors input a lot of effort to reach valuable results and proper conclusions.

Regarding structure of manuscript and content: Abstract and introduction is well written. Only in line 66, please change to [11-13] instead [11,12,13].  Results and Discussion are presented very clearly and describe all experimental results very well. Tables and figures are presented correctly. Methods and Materials are very well organize.

Comments:

Authors didn’t provided QCs in figure 1 and description is wrongly name PCA as in fact authors presented PLS-DA. Please confirm you check stability of system and reliability of results based on QC and please provide results (QCs cluster on graph)? If it is possible please provide PCA graph as well, as you mentioned about PCA investigation in the text (line 94-96).

Please confirm 2.1. title - you used "Metabonomic study" phrase. 

Please provide Figures 3, 4, 5 with better quality.

Line 294, please remove "et al" as it has been repeated

Line 288, please change to [27-28] instead [2728].

Please provide Table 1(S1) which was mentioned in the line 349

Line 356 please correct the character "ul"

Line 427 - please provide Table 1S. Is it the same as 1(S1)??

Author Response

Dear  Editor of Metabolites

Manuscript No. metabolites-821575

Title: Urine metabolome during parturition

Authors: Federica Gevi, Alessandra Meloni, Rossella Mereu, Veronica Lelli, Antonella Chiodo, Antonio Ragusa, Anna Maria Timperio.

First of all, we would like to thank you and the Reviewers for the positive and encouraging comments on the manuscript.

Please, find enclosed a point by point reply to each one of the Reviewers’ comments. We sincerely hope that this deeply revised version of the manuscript will succeed in meeting the Reviewers’ expectations. All changes are highlighted using track changes function in Microsoft word.

Kind regards

Prof. Timperio Anna Maria

Department of Ecological and Biological Sciences (DEB)

University of Tuscia

Largo dell’Università

01100 Viterbo, ITALY

Phone: +39 0761357100 (direct)

Fax: +39 0761357179 (dept.)

E-mail: timperio@unitus.it 

Reviewer #2

The manuscript titled "Urine metabolome during parturition" submitted to Metabolites, describes a very interesting and useful approach using metabolomics to evaluate the differences between two times of labor defined out of labor (OL) and in labor or dilating phase (IL-DP). The presented results confirm the retrospective clinical diagnosis of labor evidenced clusters of metabolites involved in labor condition able to discriminate between urine samples collected out of labor (OL) and during labor (in labor or dilating phase IL-DP).  The study seems to be properly designed and authors input a lot of effort to reach valuable results and proper conclusions.

We appreciate very much for the positive comments.

Regarding structure of manuscript and content: Abstract and introduction is well written. Only in line 66, please change to [11-13] instead [11,12,13].  Results and Discussion are presented very clearly and describe all experimental results very well. Tables and figures are presented correctly. Methods and Materials are very well organize.

Comments:

Reviewer #2 Authors didn’t provided QCs in figure 1 and description is wrongly name PCA as in fact authors presented PLS-DA. Please confirm you check stability of system and reliability of results based on QC and please provide results (QCs cluster on graph)? If it is possible please provide PCA graph as well, as you mentioned about PCA investigation in the text (line 94-96).

Author’s reply: the mistake has been correct  and a PLS-DA 3D QC analysis has been performed . The PCA was uploaded in supplementary materials.

Reviewer #2Please confirm 2.1. title - you used "Metabonomic study" phrase.

Author’s reply: the text wa corect

Reviewer #2 Please provide Figures 3, 4, 5 with better quality.

 We thank the reviewer to bring these to our attention. The images quality has been improved  and they have been uploaded in supplementary material files

Reviewer #2 Line 294, please remove "et al" as it has been repeated, Line 288, please change to [27-28] instead [2728]. Line 356 please correct the character "ul"

 Author’s reply: All changes requested has been correct.

Reviewer #2 Line 427 - please provide Table 1S. Is it the same as 1(S1)??

Reviewer #2 Please provide Table 1(S1) which was mentioned in the line 349

Author’s reply: We apologize unfortunately there was a problem in loading table 1S in the new version of manuscript we reloaded the table

Round 2

Reviewer 2 Report

Authors made corrections of manuscript according suggestions of reviewers. I would like to thank the authors for addressing my concerns. Manuscript could be accepted in present form.